# CCK Receptor Inhibition Reduces Pancreatic Tumor Fibrosis and Promotes Nanoparticle Delivery

**DOI:** 10.3390/biomedicines12051024

**Published:** 2024-05-07

**Authors:** Thomas Abraham, Michael Armold, Christopher McGovern, John F. Harms, Matthew C. Darok, Christopher Gigliotti, Bernadette Adair, Jennifer L. Gray, Deborah F. Kelly, James H. Adair, Gail L. Matters

**Affiliations:** 1Department of Neural and Behavioral Sciences, Penn State College of Medicine, P.O. Box 850, Hershey, PA 17036, USA; 2Department of Biochemistry and Molecular Biology, Penn State College of Medicine, P.O. Box 850, Hershey, PA 17036, USA; 3Department of Biological Sciences, Messiah University, One University Avenue, Mechanicsburg, PA 17055, USA; 4Department of Materials Science & Engineering, Pennsylvania State University, 407 Steidle Building, University Park, PA 16802, USA; 5N-022 Millennium Science Complex, Materials Research Institute, Pollock Road, University Park, PA 16802, USA; 6Department of Biomedical Engineering, The Center for Structural Oncology, 506 Chemical and Biomedical Engineering, Pennsylvania State University, University Park, PA 16803, USA; 7Departments of Materials Science & Engineering, Biomedical Engineering, and Pharmacology, The Pennsylvania State University, University Park, PA 16802, USA; jha3@psu.edu

**Keywords:** CCK2 receptor, pancreatic tumor cells, fibrosis, nanoparticle dispersion

## Abstract

The poor prognosis for pancreatic ductal adenocarcinoma (PDAC) patients is due in part to the highly fibrotic nature of the tumors that impedes delivery of therapeutics, including nanoparticles (NPs). Our prior studies demonstrated that proglumide, a cholecystokinin receptor (CCKR) antagonist, reduced fibrosis pervading PanIN lesions in mice. Here, we further detail how the reduced fibrosis elicited by proglumide achieves the normalization of the desmoplastic tumor microenvironment (TME) and improves nanoparticle uptake. One week following the orthotopic injection of PDAC cells, mice were randomized to normal or proglumide-treated water for 3–6 weeks. Tumors were analyzed ex vivo for fibrosis, vascularity, stellate cell activation, vascular patency, and nanoparticle distribution. The histological staining and three-dimensional imaging of tumors each indicated a reduction in stromal collagen in proglumide-treated mice. Proglumide treatment increased tumor vascularity and decreased the activation of cancer-associated fibroblasts (CAFs). Additionally, PANC-1 cells with the shRNA-mediated knockdown of the CCK2 receptor showed an even greater reduction in collagen, indicating the CCK2 receptors on tumor cells contribute to the desmoplastic TME. Proglumide-mediated reduction in fibrosis also led to functional changes in the TME as evidenced by the enhanced intra-tumoral distribution of small (<12 nm) Rhodamine-loaded nanoparticles. The documented in vivo, tumor cell-intrinsic anti-fibrotic effects of CCK2R blockade in both an immunocompetent syngeneic murine PDAC model as well as a human PDAC xenograft model demonstrates that CCK2R antagonists, such as proglumide, can improve the delivery of nano-encapsulated therapeutics or imaging agents to pancreatic tumors.

## 1. Introduction

Pancreatic ductal adenocarcinoma (PDAC) presents significant therapeutic challenges. PDAC patients generally are diagnosed after the development of a substantial tumor burden, including metastases. Surgical resection options are limited and resistance to standard chemotherapies is common. A characteristic feature of PDAC is the highly fibrotic tumor microenvironment (TME). The fibrotic PDAC stroma develops through the deposition of extracellular matrix (ECM) proteins that block the infiltration of immune cells, suppress tumor vascularity, and promote tumor hypoxia and metabolic reprogramming. As such, PDAC is considered to be one of the most poorly vascularized and hypoxic tumor types [1]. The fibro-inflammatory tumor microenvironment in PDAC is more pronounced than in most other solid tumors and generates locally high interstitial pressures as well as a physical barrier to the diffusion of therapeutics [2,3,4]. It has also been postulated that invasive tumor cells use the aligned stromal ECM as a migration pathway, promoting metastatic spread [5]. 

There have been some seemingly conflicting studies over the role of stroma in PDAC progression. While patients with highly desmoplastic tumors have decreased overall survival compared to patients with less fibrotic tumors [6], other studies have suggested that the ablation of the stromal fibroblasts leads to more invasive tumor growth and metastasis [7]. As such, there is strong evidence that normalizing, but not eliminating, stromal barriers may preserve its restrictive roles while increasing drug efficacy [8,9] and nanoparticle infiltration [10,11,12]. Thus, an approach to fibrosis reduction that does not completely remove a stroma cell component but rather alters the activity of an important stromal signaling pathway may prove more successful [4,13].

Numerous studies have implicated cholecystokinin-2 receptors (CCK2R) in PDAC progression and, more recently, as essential participants in the signaling pathways that lead to stellate cell activation and increased tumor fibrosis [14]. CCK2R is a G protein-coupled receptor with a well-characterized signaling pathway in normal pancreatic acinar cells [15,16], including the ligand-mediated activation of phospholipase-C through Gαq coupling. Its ligands, gastrin and cholecystokinin (CCK), drive increased tumor growth and are often aberrantly upregulated in PDAC cells [17,18]. In addition to pancreatic tumor cells, CCK2R is expressed by pancreatic stellate cells (PSCs). Primary cultures of human and rat PSCs express both CCK receptor isoforms (CCK1R and CCK2R) and respond to CCK and gastrin by secreting collagen [19,20,21]. CCK activates rat PSCs in a fashion similar to TGF-β, an established stellate cell activator, and the blockade of CCKRs on rat PSCs with either a CCK1R-specific antagonist (L-364,718) or a CCK2R antagonist (LY288513) reduces collagen production [19]. 

CCK receptor antagonists have been candidates for treating a variety of cancers [22,23]. Using LSL-Kras^G12D/+^/Pdx-Cre (KC) mice which spontaneously develop PanIN lesions with high frequency, our research team previously demonstrated that the blockade of CCK receptors with the water-soluble and orally bioavailable antagonist, proglumide, reduced the fibrosis surrounding early stage, pre-cancerous PanIN lesions [24]. The treatment of established murine PDAC with CCK receptor antagonists, including YM022 and proglumide, enhanced immune cell infiltration but did not reduce tumor mass unless animals were fed a high-fat diet [14,25]. We hypothesize that the abrogation of CCK2R signaling in tumor cells will promote the normalization of the tumor stroma, decrease tumor fibrosis, and improve tumor vascularity which, together, will positively affect nanoparticle access and perfusion. 

This work further details how the pharmacological blockade of CCK2R signaling alters the tumor microenvironment and demonstrates the effect of proglumide treatment on perfusion using the delivery of a nano-encapsulated imaging agent. The KC mice previously utilized to show the proglumide-mediated decrease in PanIN fibrosis rarely develop full PDAC during the timeframe assessed. In the project reported herein, the ability of proglumide to decrease tumor fibrosis was tested using both an orthotopic human xenograft model and an immune-competent murine PDAC tumor model. Additionally, our work expands on previous quantification that utilized differential histological staining (Masson’s trichrome) through imaging fibrosis three-dimensionally in significantly thicker portions of tumor using multi-photon, second harmonic generation (SHG) microscopy. SHG offers significantly enhanced tissue imaging depths (~1 mm) and permits the highly sensitive and quantitative direct visualization of fibrillar collagens in intact tumors without the use of exogenous probes, histological sectioning, or staining [26]. Finally, because proglumide is a broad antagonist of both CCK1 and CCK2 receptors and is nonspecifically delivered throughout the animal, we employed the stable genetic downregulation of CCK2R in PDAC tumor cells to specifically address the cell-intrinsic role of this receptor isoform in the development of tumor fibrosis.

## 2. Materials and Methods

### 2.1. Cultured Cell Lines

PANC-1 and Panc02 cell lines (from ATCC) were cultured in DMEM containing 10% FBS. Cells were verified yearly by ATCC STR authentication testing. The characterization of PANC-1 cells stably transfected with a CCK2R shRNA (sh1413) was previously reported [27].

### 2.2. In Vivo Tumor Xenografts

All animal procedures were approved by the Pennsylvania State University and Messiah University Institutional Animal Care and Use Committees (IACUCs). The PSU College of Medicine Animal Resource Program is accredited by the Association for Assessment and Accreditation of Laboratory Care International (AAALAC International). All animals’ living conditions were consistent with the standards required by AAALAC International. Four- to six-week-old male athymic (nu/nu) mice were purchased from Charles River. Each xenograft experiment had at least 6 mice per treatment group and was replicated at least twice for rigor and reproducibility. To establish orthotopic pancreatic cancer xenografts, mice were fully anesthetized, a small incision was made in the left flank, the peritoneum was dissected, and the pancreas was exposed. Tumor cells (1 × 10^6^ cells in 100 µL of Hank’s balanced salt solution; HBSS) were injected into the pancreas, and the surgical site was closed with staples. 

### 2.3. Syngeneic Tumor Growth

Four- to six-week-old male C57Bl/6 mice (Charles River; Wilmington, MA, USA, Taconic, Germantown, NY, USA) underwent intrapancreatic injection of Panc02 cells (1 × 10^6^ cells prepared in 50 or 100 µL HBSS) as described above. Each xenograft experiment had at least 6 mice per treatment group and was replicated at least twice for rigor and reproducibility.

### 2.4. Proglumide Treatment

Proglumide sodium salt (4-benzamido-5-(dipropylamino)-5-oxopentanoic acid; Sigma, St. Louis, MO, USA; Tocris Bioscience, Minneapolis, MN, USA) was prepared weekly in normal drinking water. One week after tumor cells were implanted into the pancreas, animals were placed on either proglumide-treated drinking water (0.1 mg/mL provided ad libitum) or normal drinking water (vehicle control) for 4–6 weeks [24]. 

### 2.5. Ex Vivo Multiphoton Microscopy/Second Harmonic Generation (SHG) Tumor Imaging

Multiphoton microscopy, which uses ultra-short IR laser pulses as the excitation source, produces multiphoton excitation fluorescence signals from exogenous or endogenous fluorescent proteins and induces specific second harmonic generation (SHG) signals from non-centrosymmetric proteins such as fibrillar collagens in ex vivo tissues. Prior to euthanasia, mice were injected (intravenously) with tomato lectin–FITC (Sigma L0401), and after 10 min, blood was cleared by intracardiac perfusion with cold, heparinized saline (1 mL/min). Pancreata were fixed in 4% PFA (24 h at 4 °C), cryo-protected in 10% sucrose (overnight at 4 °C), and transferred to 30% sucrose. Cryo-protected tissues were subsequently frozen in OCT and imaged using a Nikon A1 MP+ multi-photon microscope system to assess the location and quantity of tumor collagen. The laser was a mode-locked, femto-second, single-box laser system with automated dispersion compensation (Spectra-Physics, Andover, MA, USA). The analysis of multidimensional images was performed using an IMARIS/VOLOCITY image processing workstation.

### 2.6. Fibrosis Staining and Immunohistochemical (IHC) Staining

Tumor tissues fixed in PFA were paraffin-embedded, sectioned, and stained with hematoxylin and eosin (H–E) followed by fibrosis staining with Masson’s Trichrome as described [24]. Photomicrographs of Masson’s trichrome-stained tumors were analyzed using ImageJ v.1.54i processing software to quantify the proportion of blue stain collagen tissue as a percentage of the total cross-sectional area. Immunohistochemistry of serial sections was performed with antibodies to collagen I (abcam34710, 1:100), alpha smooth muscle actin (abcam15734 1:100), or CD31 (abcam28364, 1:50). Staining was visualized using an ImmPRESS Excel Staining Kit (MP-7601, Vector Labs, Newark, CA, USA) with a one-minute DAB incubation and documented with a BZ-X710 All-in-one fluorescence microscope (Keyence, Elwood Park, NJ, USA) under bright field conditions. Quantitative IHC image analysis was also performed using ImageJ software, and at least 10 fields per slide were analyzed.

### 2.7. Characterization and Imaging of Rhodamine NanoJackets (RhodNJs)

Rhodamine NanoJackets (RhodNJs), synthetic, nano-sized particles composed of calcium and phosphate that are non-toxic and stable in circulation [28], were prepared via a microemulsion technique as previously described [29]. Rhodamine WT encapsulation was accomplished through the addition of the fluorophore into the microemulsion phase such that the fluorescent molecules were trapped and internalized within the particle [30]. To quantify rhodamine encapsulation, particles were dissolved to release the dye, and fluorophore content was quantified by the optical absorbance and compared to a standard curve. 

To verify the size and shape of the RhodNJs, particles were imaged using liquid cell scanning transmission electron microscopy with the HAADF detector (HAADF-STEM) on a Thermo Fisher Talos F200X (Waltham, MA, USA) electron microscope (Appendix A) [31]. Normal STEM conditions were used, including spot size 7, which provides a current of approximately 0.1 nA. The size distribution of the NJs was determined using ImageJ software (*n* = 134 nanoparticles evaluated in 6 different liquid cell STEM photomicrographs). Data were deconvoluted with PeakFit v2.14 (Systat) to generate statistical analyses of the percent area (a_0_), central value (a_1_), and standard deviation (a_2_) of the peaks. There was no correction made for baseline.

For animal studies, RhodNJs were diluted into sterile, phosphate-buffered saline, pH 7.4, and injected via the lateral tail vein into tumor-bearing mice. At 24 h after RhodNJ injection, mice were perfused with an endothelial cell stain, tomato lectin-FITC, followed by a cold saline wash, tumor excision, and cryo-protection as described above. The 24 h post-injection timepoint was selected based on prior studies that suggested the majority of NJs not taken up by tumors are cleared within 18–24 h [32,33]. After ex vivo fixation and sectioning, a Nikon A1 MP+ multi-photon microscope system was again used to assess localization of the Rhodamine signal relative to regions of the tumor endothelium [33].

### 2.8. Statistical Analyses

Data are presented as mean ± standard error of the mean or as mean ± 95% confidence interval as denoted in the figure legends. Comparison of the means between groups was carried out using unpaired two-tailed *t*-tests or one-way ANOVA with Prism 6.0 software (GraphPad), and *p* < 0.05 was considered significant. For nanoparticle size analysis, PeakFit v.4.12 (Systat) was used to deconvolute the 4 peaks from the experimental data and generate the statistical analyses of the percent of area (a_0_), central value (a_1_), and standard deviation (a_2_). There was no correction made for baseline.

## 3. Results and Discussion

### 3.1. Three-Dimensional Ex Vivo Imaging Demonstrates Proglumide Reduces Fibrillar Collagen

To assess proglumide-mediated fibrotic inhibition in a model of established PDAC, orthotopic human PANC-1 tumors were established in athymic mice and animals were placed on normal or proglumide-treated drinking water one week following tumor cell injection. Following 4 weeks of tumor growth, the ex vivo imaging of 1–3 mm tumor sections was performed with multi-photon, second harmonic generation (SHG) microscopy. Orthotopic PANC-1 tumors from proglumide-treated mice showed significantly less fibrillar collagen than tumors from mice receiving normal drinking water (Figure 1A,B, *** *p* < 0.0001). Importantly, although proglumide decreased the collagen signal by approximately 53% (Figure 1B), this dose of proglumide did not completely eliminate tumor fibrosis. 

### 3.2. Histopathology Confirms Decreased Fibrosis and Reprogrammed Tumor Stroma

To verify the reduction in tumor fibrosis in proglumide-treated mice, paraffin-embedded portions of PANC-1 tumor were sectioned and stained with Masson’s trichrome. Similar to what was seen in the three-dimensional imaging, trichrome-stained tumor sections showed a 31% reduction in tumor collagen (blue stain) in proglumide-treated mice (Figure 2A,B). Somewhat less than the 53% reduction assessed by SHG, this could be due in part to the differences in the types of ECM proteins and fibrillar collagen (visualized by SHG) versus total collagen (visualized by trichrome) measured by these two methods [34]. Nonetheless, the data are consistent with the overall conclusion that proglumide reduces pancreatic tumor fibrosis in an in vivo orthotopic tumor xenograft model. In a third parallel approach, PANC-1 orthotopic tumor sections were subjected to immunohistochemical (IHC) staining with anti-collagen1 antibodies. Type 1 collagen was shown again to be reduced in the proglumide-treated tissue compared with the vehicle control (Figure 2D). The quantitation of IHC staining revealed a 37% decrease in collagen 1 in tumors from proglumide-treated mice, in agreement with the other methods used for collagen/ECM quantification. Additionally, all three visualization methods suggested that the alignment of the collagen fibers in tumors from proglumide-treated mice appeared less linearly aligned, indicative of a reprogrammed PDAC tumor stroma [35]. Finally, since much tumor collagen is believed to be produced by activated stellate cells/CAFs, sections were stained for αSMA, a marker of activated stellate cells and the myofibroblastic CAFs (myCAFs) implicated in tumor progression [36]. Proglumide reduced the number of αSMA-expressing cells by 23% compared with vehicle controls (Figure 2E).

### 3.3. Proglumide Reduces Tumor Fibrosis in an Immunocompetent Pancreatic Cancer Model

To confirm the efficacy of proglumide in an immunocompetent mouse model of PDAC, orthotopic Panc02 murine pancreatic cancer tumors were established in syngeneic C57Bl/6 mice. Animals were provided either normal or proglumide-treated drinking water. Water consumption was tracked daily for groups of 1–3 mice, demonstrating the delivery of 15–20 mg/kg/day of proglumide. At necropsy, tumor mass in treated animals (0.298 g ± 0.082 StErr) was unchanged from control animals (0.284 g ± 0.053). Tumor fibrosis was analyzed ex vivo by both SHG (Figure 3A) and Masson’s trichrome (Figure 3B). Although the development of fibrosis was slightly more variable in this model than in human PANC-1 tumors, both stromal evaluation methods showed a 33% decrease in the fibrotic tumor microenvironment in proglumide-treated animals, confirming other studies using proglumide and the Panc02 cell line [25].

### 3.4. Tumor Cell CCK2 Receptor Drives Pancreatic Tumor Fibrosis

To distinguish whether the anti-fibrotic effect of proglumide is mediated through CCK2 receptors expressed on tumor cells or through CCK2Rs expressed on other TME cells, such as pancreatic stellate cells or other tumor fibroblasts [19], we utilized PANC-1 clones in which the CCK2 receptor expression had been stably knocked down with a gene-specific shRNA (sh1413) [27]. PANC-1 cells with CCK2R shRNA knock-down were then orthotopically implanted into athymic mice that expressed normal levels of CCK2R in their TME cells. When compared to tumors established with parental (CCK2R expressing) PANC-1 cells, fibrosis in tumors established with PANC-1 cells with CCK2R down-regulation was significantly reduced (Figure 4). Unlike tumors grown in proglumide-treated mice, the stable shRNA knockdown of CCK2R resulted in a greater reduction in pancreatic tumor fibrosis of nearly 75% (Figure 4A). This four-fold decrease in total fibrosis indicates that CCK2R signaling within the tumor cells themselves contributes to the development of TME desmoplasia. Demonstrating that genetically abrogating the CCK2 receptor in tumor cells themselves resulted in a greater reduction in whole tumor collagen content suggests that tumor cell-intrinsic CCK2R signaling is a critical factor in the development of fibrosis in pancreatic cancer. 

### 3.5. Fibrosis Reduction from Proglumide Treatment Improves Vascularity, Perfusion, and Delivery of Imaging Nanoparticles to the Tumor Microenvironment

The densely packed ECM of the PDAC microenvironment decreases vascular patency, limits drug efflux, and prompts metabolic reprogramming [37]. The increased stiffness of the desmoplastic stroma of pancreatic tumors compresses tumor vessels, resulting in impaired tumor perfusion. Three quarters of vessels in pancreatic tumors show signs of vascular collapse [38]. Elevated intra-tumoral hydrostatic pressure impairs bulk exchange from the vasculature and diffusion of therapeutics, and the resulting hypoxia also promotes immunosuppression, disease progression, and treatment resistance [39]. Indeed, others have demonstrated that collagen density and intra-tumoral pressure are directly correlated, and fibrosis reduction strategies improve the delivery of small molecules, such as nanoparticles, to tumors [40,41,42,43]. To functionally assess the impact of proglumide treatment on tumor vascularity and nanoparticle dispersion, orthotopic PANC-1 tumors were again established in athymic mice. One week after tumor implantation, half of the mice were placed on proglumide-treated water for 4 weeks and the reduction in tumor collagen in proglumide-treated mice was confirmed by SHG (Figure 5A). The IHC quantification of CD31 in tumor sections from proglumide-treated mice revealed a significant increase in CD31 immunostaining (Figure 5B), which is indicative of increased vascularity. 

However, CD31 staining by itself does not demonstrate a change in the functionality of tumor vessels. To achieve this, we evaluated the dispersion and uptake of imaging nanoparticles in both treated and untreated tumors. Calcium phosphosilicate nanoparticles formulated to encapsulate rhodamine WT, referred to as Rhodamine NanoJackets (RhodNJs), were generated [29,44,45]. RhodNJs can encapsulate and deliver imaging agents and are easily detected, localized, and quantified in tumors ex vivo. Prior to injection, the characterization of RhodNJs was performed via liquid cell scanning transmission electron microscopy (STEM), and the STEM images were used to determine mean particle diameter and particle size distribution. Contrast is reversed with respect to STEM so that nanoparticles appear brighter in the sample (Figure 5C). Based on STEM micrographs (*n* = 134 particles), the size distribution of the RhodNJs was determined, which demonstrated that the NJs had a mean diameter of 12 nm (Figure 5D). Since the majority of the nanoparticles (>70%) were 12 nm or less, they were in the ideal size range to assess the diffusion capacity of the tumor vasculature in proglumide-treated versus untreated mice.

PANC-1 tumor-bearing mice again were placed on either normal or proglumide-treated water for four weeks, and Rhodamine NanoJackets (RhodNJs) were administered via a lateral tail vein at 24 h prior to necropsy. Mice were also perfused with tomato lectin–FITC 10 min prior to euthanasia followed by a cold saline wash, so that the location of the vasculature beds within the tumor could be visualized. In comparison to tumors from untreated mice, tumors from proglumide-treated mice had a significantly stronger overall rhodamine nanoparticle signal (Figure 5E). To illustrate the effective dispersal of nanoparticles into surrounding tumor tissue in proglumide-treated mice, the location of the rhodamine nanoparticles was compared to the location of the tumor blood vessels tagged with the endothelial cell lectin stain (green signal, Figure 5F). A merged image of the rhodamine and endothelial signals demonstrated that although some of the RhodNJs co-localized with the tumor vessels at 24 h after intravenous injection (Figure 5F, yellow on merged image), many of the nanoparticles were present in the tumor matrix outside of the blood vessels (Figure 5F, red on merged image). Prior work by our team has shown that NJs are stable in vivo for up to 36 h and that NJ cargo is only released in a pH-dependent manner after uptake by tumor cells and internalization in lysosomes. Thus, it is unlikely that the signal represents free rhodamine released prior to vascular extravasation [33]. Together, these data suggest that the proglumide-mediated reduction in collagen altered tumor vascularity, contributed to increased tumoral perfusion, and enhanced the delivery of the RhodNJ imaging nanoparticles to tumors.

## 4. Conclusions

A desmoplastic tumor microenvironment is a hallmark of pancreatic cancer [3]. Neoplastic cells make up only a small portion of PDAC tumors, while stromal components such as cancer-associated fibroblasts (CAFs), immune cells, extracellular matrix proteins, and hyaluronan typically make up the bulk of the tumor volume [46]. In pre-clinical PDAC models, it has been shown that the complete elimination of stromal fibroblasts by genetic and/or pharmacological techniques produces a more aggressive disease and shorter survival [7,47]. Since ECM can have a protective effect in restraining tumor growth and progression, approaches to normalize or re-educate, but not eliminate, stromal components could be more effective.

We have documented a significant anti-fibrotic effect of proglumide in vivo using murine PDAC in both immunocompetent transgenic [24] and syngeneic models (Figure 3), as well as a human PDAC xenograft model (Figure 1). In addition to the traditional histological quantification of total collagen by Masson’s trichrome (Figure 2C), we have confirmed a specific reduction in Type 1 collagen via IHC (Figure 2D). Unlike prior studies, the SHG analysis (Figure 1 and Figure 3) permits the quantification of fibrillar collagen in larger three-dimensional tumor portions less susceptible to the sampling biases of traditional histological sectioning and is inherently more quantitative than the image analysis of photomicrographs based on pixel color. Using multiple models and analytic approaches, we have consistently demonstrated a decrease in pancreatic tumor fibrosis in animals receiving proglumide.

This study indicates that proglumide contributes to the reprogramming of the TME based on four significant changes in tumor matrix structure and cellular composition. First, histology showed a significant increase in the number of CD31-positive endothelial cells after proglumide treatment, denoting an increase in vascularity. Second, evidence that CD31-positive vascular structures in the proglumide-treated tumors were more linear and elongated than in untreated mice suggests a decrease in the abnormal vasculature that is characteristic of aggressive pancreatic carcinomas [35,39]. Third, lectin staining and nanoparticle detection demonstrated the functional patency of the vessels for improved nanoparticle delivery. Fourth, a decrease in αSMA-positive cells in proglumide-treated tumors suggests either less activation of, or a fewer number of, the PSCs and myCAFs, which is emblematic of a reactive, desmoplastic microenvironment [36].

The pharmacological blockade of CCK2 receptors may be an effective new therapeutic approach for blocking tumor-stellate cell communication and rewiring the TME through normalized human tumor vasculature and reduced stellate cell activation. Communication among the tumor cells and multiple stromal cell types of the desmoplastic TME is complex [48], and the exact cellular target(s) of proglumide are currently unclear. Our work, however, provides mechanistic insights. The four-fold decrease in total fibrosis in orthotopic tumors derived from human PANC-1 cells with the shRNA-mediated genetic knockdown of CCK2R indicates that CCK2R signaling within the tumor cells themselves contributes to the development of TME desmoplasia. There could be several mechanisms by which proglumide could act. First, as PDAC tumor cells are known to synthesize collagen [49], the blockade of CCK2R by proglumide could directly reduce collagen secretion by PDAC cells. Because of the autocrine stimulation of CCK2R by PDAC-secreted gastrin [17], competition between tumor gastrin and the reversible antagonist proglumide also may explain the greater fibrotic inhibition achieved by receptor downregulation. Second, proglumide could abrogate signaling through tumor cell CCK2R that triggers downstream paracrine pathways to activate CAFs/PSCs. Finally, it is also possible that proglumide could block CCK2R signaling in both tumor and stromal compartments and that tumor cell CCK2R signaling potentially impacts the relative abundance of tumor-suppressive versus tumor-promoting CAF subpopulations within the stroma.

Many studies have shown that the desmoplastic stroma of pancreatic tumors increases the stiffness of the tumor matrix and compresses tumor vessels resulting in impaired tumor perfusion. Others have shown that collagenase injections into established orthotopic human PDAC tumors led to decreased intra-tumoral pressure and increased drug perfusion [50]. More recently, a 13% reduction in PDAC tumor collagen was observed after sub-lethal photodynamic therapy, also known as photodynamic priming [51]. It has been shown that this photo-priming also results in the better delivery of adjuvant therapies [52]. Other drugs, such as ACE inhibitors, have been used to improve tumor perfusion and enlarge endothelial gaps and enhance nanomedicine delivery to tumors [41]. Proglumide, which non-invasively targets the collagen component of the TME, was recently shown to increase the efficacy of chemotherapy as part of a combination therapeutic [40]. Our study provides further functional evidence that proglumide can also be an effective partner in the delivery of nanomedicines by improving the ability of small nanoparticles to accumulate in PDAC tumors. New nanomaterials are being developed for the treatment of PDAC including immune-modulating nanoparticles [53]. In pancreatic tumors, blood vessel pores are in the range of 50–60 nm in diameter [54], and the delivery of targeted nanoparticles occurs in a size-dependent manner [11]. Normalization of tumor vasculature that improves vessel patency and decreases the interstitial fluid pressure in tumors should allow these small nanoparticles to enter tumors more efficiently. The NanoJackets used in this study, with an average diameter of ~12 nm (Figure 5D), are well suited to assess changes in tumor vasculature in comparison to larger nanoparticles such as liposomes [11]. In the future, the combination of small nanoparticles or small molecule therapeutics with proglumide could more effectively deliver drugs, immunotherapies, or imaging agents to pancreatic tumors. 

Together, these preclinical studies provide a strong rationale for targeting the collagenous facet of PDAC fibrosis via the normalization of the desmoplastic TME. Since proglumide improved tumor perfusion, it represents an attractive and promising candidate as a combinatorial therapeutic. While we have now documented the anti-fibrotic effects of proglumide in vivo using syngeneic and xenograft models, it will be of interest to learn whether the TME-modifying functions of proglumide can reverse established fibrosis in the advanced tumor stages at which PDAC patients typically present and if proglumide would have similar effects in other tumor types. While this study has demonstrated that CCK2 receptors residing on the tumor cells are a major contributor to tumor fibrosis, the further elucidation of downstream signaling pathways and potential paracrine crosstalk within the TME will be important for mechanistically defining the proglumide-mediated stromal reprogramming that permits the improved vascularity, perfusion, and nanoparticle delivery.

## Figures and Tables

**Figure 1 biomedicines-12-01024-f001:**
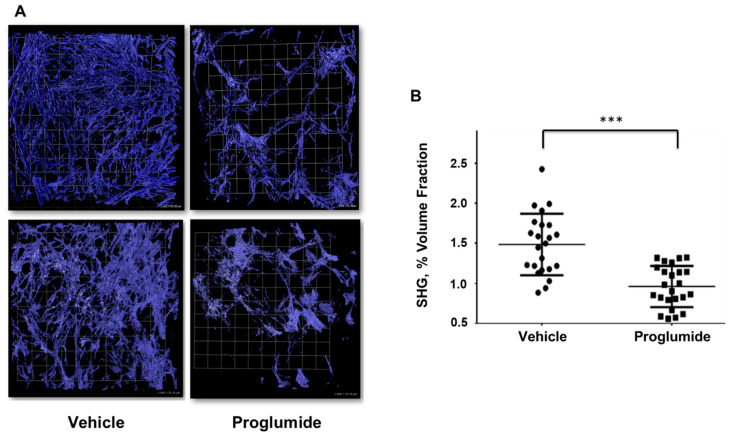
Pancreatic tumor collagen content is reduced by proglumide treatment. (**A**) After four weeks of proglumide treatment, ex vivo tumor SHG analysis of fibrillar collagen in PANC-1 tumors from mice that received vehicle (normal drinking water) or proglumide-treated drinking water showed less fibrillar collagen in tumors from the proglumide-treated mice. Scale bar box = 51.55 µm. (**B**) SHG quantitation revealed a 53% reduction in collagen content with proglumide treatment; mean ± 95% confidence interval, *** *p* < 0.0001.

**Figure 2 biomedicines-12-01024-f002:**
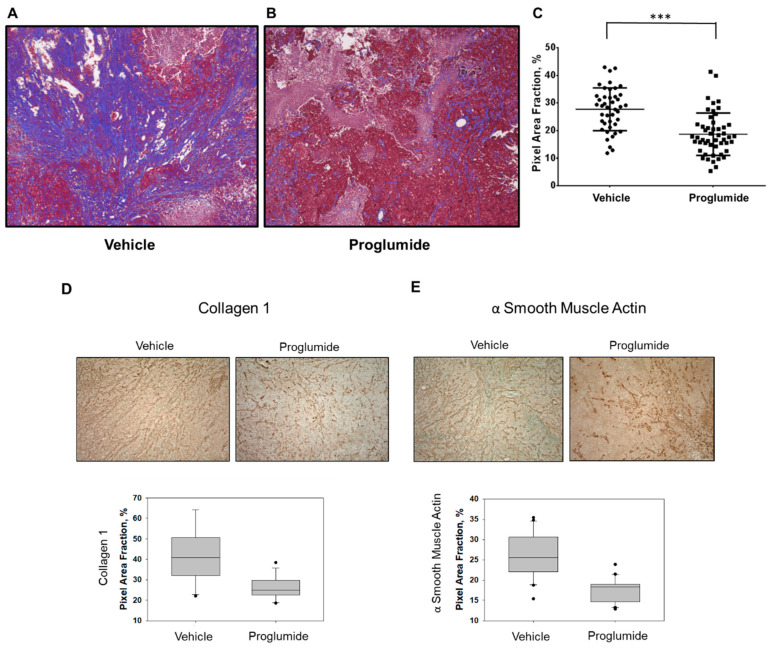
Immunohistological and trichrome staining verifies the reduction in tumor fibrosis in proglumide-treated mice compared to vehicle. (**A**) Representative images of Masson’s trichrome-stained PANC-1 tumor sections from tumor-bearing mice given normal drinking water or (**B**) proglumide-treated drinking water. (**C**) Quantitative analysis of trichrome fibrosis staining in vehicle and proglumide-treated mice demonstrated that proglumide treatment reduced collagen fibers by 31%. (*** *p* < 0.001). (**D**) Immunohistochemical staining of tumor sections reveals a reduction in Type 1 collagen in the proglumide-treated tissue compared with the vehicle control (*p* = 0.0002) as well as reduced numbers of α smooth muscle actin-positive cells in the proglumide-treated tissue compared with the vehicle control (**E**) (*p* < 0.0001) (scale bars unavailable).

**Figure 3 biomedicines-12-01024-f003:**
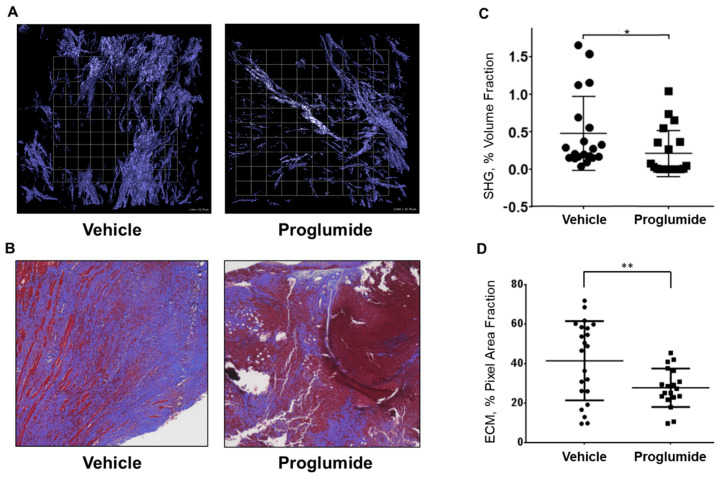
Fibrosis in murine Panc02 tumors was reduced in proglumide-treated mice. To confirm the efficacy of proglumide in an immunocompetent mouse model of PDAC, Panc02 murine pancreatic cancer tumors were established in syngeneic C57Bl/6 mice. Animals were provided either normal (vehicle) or proglumide-treated drinking water. Fibrosis reduction in proglumide-treated mice compared to untreated mice as determined by SHG (**A**) and by Masson’s trichrome (**B**) is similar in scope to the reduction in the PANC-1 model, with an average reduction in tumor fibrosis of 33% in proglumide-treated mice (**C**,**D**). SHG scale bar box = 25.78 µm; * *p* < 0.05, ** *p* < 0.005.

**Figure 4 biomedicines-12-01024-f004:**
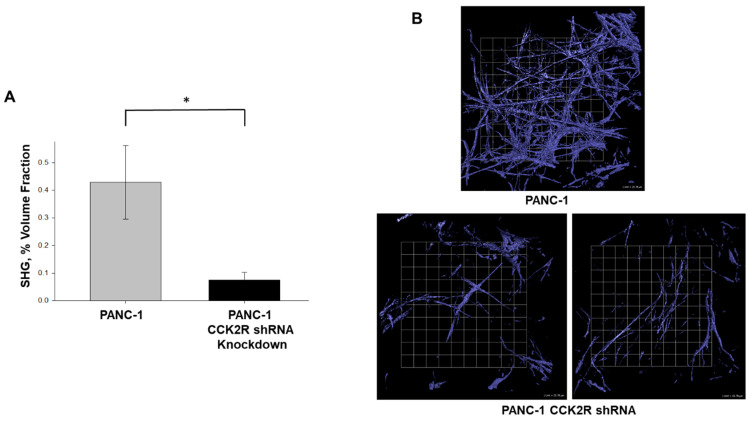
Knockdown of the CCK2 receptor in PANC-1 pancreatic cancer cells decreased tumor fibrosis in vivo. (**A**) Tumors were established in athymic mice using parental PANC-1 cells or PANC-1 cells with stable shRNA-mediated knockdown of the CCK2 receptor. SHG quantification demonstrates a four-fold reduction in fibrillar collagen in tumors with CCK2 receptor downregulation (* *p* < 0.05). (**B**) Representative SHG images emphasize the profound decrease in tumor fibrosis in CCK2-downregulated tumors. Scale bar box = 25.78 μm.

**Figure 5 biomedicines-12-01024-f005:**
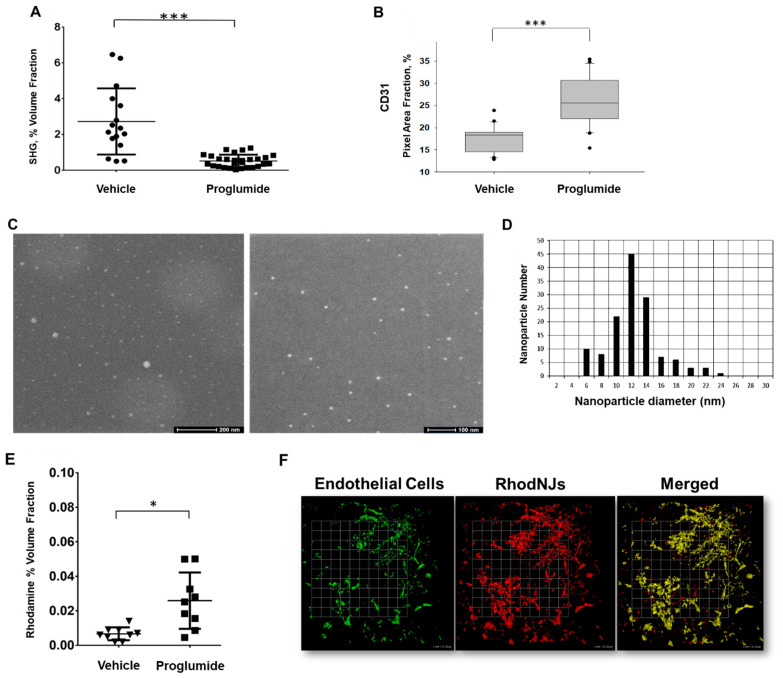
Proglumide-enhanced tumor vascularity and nanoparticle accumulation within pancreatic tumors. (**A**) Mice bearing PANC-1 tumors were provided normal or proglumide-treated drinking water for 4 weeks. A reduction in tumor collagen in proglumide-treated mice was confirmed via SHG. (**B**) CD31 immunostaining of tumor sections shows an increased tumor vascularity in proglumide-treated mice (*** *p* < 0.0001). (**C**) To functionally assess tumor vasculature status and nanoparticle diffusion in vivo, mice were administered Rhodamine NanoJackets (RhodNJs) via a lateral tail vein at 24 h prior to necropsy. Prior to injection, characterization of RhodNJs was performed via liquid cell scanning transmission electron microscopy (STEM), and the STEM images were used to determine mean particle diameter and particle size distribution. Contrast is reversed with respect to STEM so that nanoparticles appear brighter in the sample. Micrograph size bars are 200 nm (left) and 100 nm (right), respectively. (**D**) Size distribution of the RhodNJs demonstrates that RhodNJs have a mean diameter of 12 nm. The majority of the nanoparticles (>70%) were 12 nm or less, making them capable of diffusing out of the tumor vasculature (*n* = 134 particles). (**E**) At 24 h after RhodNJ injection, mice were perfused with tomato lectin–FITC concurrent with exsanguination. Scatterplot of tumoral rhodamine signal indicates that the RhodNJs accumulated to a greater degree in tumors of proglumide-treated mice (*n* = 4 mice per group; 4 images per tumor; * *p* < 0.05). (**F**) Ex vivo localization of RhodNJs (red) relative to endothelial cells (green) in a proglumide-treated tumor suggests that while some RhodNJs remain within the tumor vasculature (yellow), there are multiple areas where RhodNJs have exited the tumor vasculature (red on the merged image), scale bar box = 51.32 μm.

## Data Availability

The original contributions presented in the study are included in the article/Appendix A, further inquiries can be directed to the corresponding author.

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
