# Peer review of "CCK Receptor Inhibition Reduces Pancreatic Tumor Fibrosis and Promotes Nanoparticle Delivery"

_biomedicines, 2024, doi:10.3390/biomedicines12051024_

Round 1
Reviewer 1 Report
Comments and Suggestions for Authors
The authors have investigated the therapeutic efficacy of targeting CCK2R signaling. Althoug the study is well designed, but few more things would strengthen the findings
1. The authors showed that proglumide reduces fibrosis and tumor stroma, it would be better to establish the anti-tumor effects in subcutaneous models and add the add the data for tumor size and weight.
2. The authors showed the tumorigenicity in CCK2R KD cells in-vivo. However, they should add the data for proliferation, multiplicity and colonization.
3. The authors have not described the signaling axis as such. How is proglumide modulating the signaling axis?
4. CCK2R receptor expression is restricted to tumor cells or CAFs and how does KD of the receptor from tumor cells solve the problem ? Please discuss in the discussion.
5. Please mention the n in the animal experiments.
6. The orthotopic tumor weight can provide additional data for extent of stroma.
Author Response
Reviewer #1 Comments and Responses:
The authors have investigated the therapeutic efficacy of targeting CCK2R signaling. Although the study is well designed, but few more things would strengthen the findings
- The authors showed that proglumide reduces fibrosis and tumor stroma, it would be better to establish the anti-tumor effects in subcutaneous models and add the add the data for tumor size and weight.
We agree these are important experiments. We have recently completed a subcutaneous experiment with and without proglumide. Although the tumor size and weight data has been collected, the effect of proglumide on fibrosis in the subcutaneous tumor location is on-going and not yet completed.
We feel this paper focuses on orthotopic effects of proglumide and would prefer not to delay this publication while we complete the subcutaneous experiments. In addition, we have completed only 1 in vivo replicate subcutaneously and would require at least another replicate for rigor and reproducibility and to be publishable.
- The authors showed the tumorigenicity in CCK2R KD cells in-vivo. However, they should add the data for proliferation, multiplicity and colonization.
Characterization, including data for the proliferation, of the CCK2R KD clones is included in a prior paper by our team:
Downregulation of the CCK-B receptor in pancreatic cancer cells blocks proliferation and promotes apoptosis. Fino KK, Matters GL, McGovern CO, Gilius EL, Smith JP. Am J Physiol Gastrointest Liver Physiol. 2012 Jun 1;302(11):G1244-52. PMID: 22442157.
However, this was the first time that the CCK2R KD clone was assessed for tumor fibrosis.
- The authors have not described the signaling axis as such. How is proglumide modulating the signaling axis?
Again, an excellent suggestion. We have begun work on possible signaling read-outs for the plus and minus proglumide treatments at the RNA level, but have not yet seen differences in gene expression for a variety of downstream effectors. The potential read-outs from GPCRs such as CCK2R are numerous, and we continue to work to identify a specific signaling mechanism affected by proglumide. As these are on-going experiments, we plan to continue these analyses and to follow up on this in a future publication.
- CCK2R receptor expression is restricted to tumor cells or CAFs and how does KD of the receptor from tumor cells solve the problem ? Please discuss in the discussion.
This is a key point we wanted to make – although proglumide may act on either the tumor cell CCK2 receptor or CAF CCK2 receptor, or likely both, the tumor cell CCK2R KD experiment suggests that the tumor cell CCK2 receptor is critical to the development tumor fibrosis independent of CAFs. See lines 402 to 415 in the conclusion section where we addressed this. We are happy to rewrite or expand upon this if necessary. Our apologies if this was unclear.
- Please mention the n in the animal experiments.
Thank-you. This has been added to the methods section, Lines 125-126 and 135-136.
- The orthotopic tumor weight can provide additional data for extent of stroma.
Tumor weight data has been added to the text. There was no significant difference between the weight of tumors from control animals versus tumors from proglumide treated animals (lines 260-261).
Reviewer 2 Report
Comments and Suggestions for Authors
The authors presented interesting research on CCK receptor inhibition to improve nanoparticle delivery by reducing pancreatic tumor fibrosis. The study is quite interesting and reveals novel findings in this field. The results were well-presented and appropriately discussed. The methods also provide sufficient information for the study. However, I have a few concerns that need to be addressed before publication.
- Does this study apply only to nanoparticle-based drug delivery, or does it also hold true for drug delivery without nanoparticles?
- The authors only tested one type of nanoparticle, but the internalization of nanoparticles can depend on their functionalization. Did the authors investigate the effect of nanoparticle functionalization? Do they have any plans to do so?
- Did the authors quantify the rhodamine in the tumor using LC-MS or another quantification method besides microscopy? If not, why not?
- The study could have been conducted using rhodamine-loaded nanoparticles and rhodamine alone to determine whether the nanoparticles themselves are being internalized or if only the released rhodamine is being internalized. How did the authors determine that the nanoparticles, and not just the released rhodamine, were being internalized?
Author Response
Reviewer #2 Comments and responses:
The authors presented interesting research on CCK receptor inhibition to improve nanoparticle delivery by reducing pancreatic tumor fibrosis. The study is quite interesting and reveals novel findings in this field. The results were well-presented and appropriately discussed. The methods also provide sufficient information for the study. However, I have a few concerns that need to be addressed before publication.
- Does this study apply only to nanoparticle-based drug delivery, or does it also hold true for drug delivery without nanoparticles?
Yes – our colleague Dr. Smith at Georgetown University has shown that proglumide does enhance the action of gemcitabine and immunotherapy for orthotopic pancreatic and liver tumors. We recommend her papers on this topic:
Treatment with a Cholecystokinin Receptor Antagonist, Proglumide, Improves Efficacy of Immune Checkpoint Antibodies in Hepatocellular Carcinoma.
Shivapurkar N, Gay MD, He AR, Chen W, Golnazar S, Cao H, Duka T, Kallakury B, Vasudevan S, Smith JP. Int J Mol Sci. 2023 Feb 11;24(4):3625. PMID: 36835036.
Cholecystokinin Receptor Antagonist Improves Efficacy of Chemotherapy in Murine Models of Pancreatic Cancer by Altering the Tumor Microenvironment.
Malchiodi ZX, Cao H, Gay MD, Safronenka A, Bansal S, Tucker RD, Weinberg BA, Cheema A, Shivapurkar N, Smith JP. Cancers (Basel). 2021 Sep 30;13(19):4949. PMID: 34638432
- The authors only tested one type of nanoparticle, but the internalization of nanoparticles can depend on their functionalization. Did the authors investigate the effect of nanoparticle functionalization? Do they have any plans to do so?
Yes – we have previously shown that nanoparticle functionalization increases tumor up-take. Please see our previous publications:
Thomas Abraham, Christopher O McGovern, Samuel S Linton, Zachary Wilczynski, James H Adair, Gail L Matters. Aptamer-Targeted Calcium Phosphosilicate Nanoparticles for Effective Imaging of Pancreatic and Prostate Cancer. Int J Nanomedicine. 2021 Mar 19:16:2297-2309. PMID: 33776434.
Preferential uptake of antibody targeted calcium phosphosilicate nanoparticles by metastatic triple negative breast cancer cells in co-cultures of human metastatic breast cancer cells plus bone osteoblasts. Bussard KM, Gigliotti CM, Adair BM, Snyder JM, Gigliotti NT, Loc WS, Wilczynski ZR, Liu ZK, Meisel K, Zemanek C, Mastro AM, Shupp AB, McGovern C, Matters GL, Adair JH. Nanomedicine. 2021 Jun;34:102383. PMID: 33722692.
Now that we have established that proglumide enhances dispersion of a generalized, nontargeted nanoparticle, our on-going experiments are now testing the combination of proglumide with targeted nanoparticles. We are also in discussion with new collaborators to investigate whether the action of proglumide can enhance up-take of other classes of nanoparticles.
- Did the authors quantify the rhodamine in the tumor using LC-MS or another quantification method besides microscopy? If not, why not?
An excellent suggestion which we will consider in future experiments. In order to achieve the 3-D confocal imaging of tumor fibrosis by SHG, we OCT fix and use all the tumor tissue for that analysis.
In addition, the data in Figure 5 is meant to demonstrate not only enhanced up-take in proglumide treated tumors, but also the localization of the Rhodamine-NJs outside of the vascular and peri-vascular regions. LC-MS could certainly verify the increased quantity of rhodamine but not its extra-vascular localization.
We will address the Rhodamine quantification via a second method in future work. Thank-you for the suggestion.
- The study could have been conducted using rhodamine-loaded nanoparticles and rhodamine alone to determine whether the nanoparticles themselves are being internalized or if only the released rhodamine is being internalized. How did the authors determine that the nanoparticles, and not just the released rhodamine, were being internalized?
Again, an excellent point that we should have brought out in the manuscript. Prior work by our team has shown that the rhodamine NJs are stable in vivo for up to 36 hours and do not leak Rhodamine. (Thomas Abraham, Christopher O McGovern, Samuel S Linton, Zachary Wilczynski, James H Adair, Gail L Matters. Aptamer-Targeted Calcium Phosphosilicate Nanoparticles for Effective Imaging of Pancreatic and Prostate Cancer. Int J Nanomedicine. 2021 Mar 19:16:2297-2309. PMID: 33776434).
Other previous work by co-author Dr. Adair has demonstrated that NJ cargo is only released in a pH dependent manner after up-take by tumor cells and internalization in lysosomes.
Calcium phosphate-based composite nanoparticles in bioimaging and therapeutic delivery applications. Tabaković A, Kester M, Adair JH. Wiley Interdiscip Rev Nanomed Nanobiotechnol. 2012 Jan-Feb;4(1):96-112. PMID: 2196517.
We have added text to the results section to address this concern (lines 344-348).
Reviewer 3 Report
Comments and Suggestions for Authors
Journal Title: Biomedicines
Manuscript ID: biomedicines-2959503
Title: CCK receptor inhibition reduces pancreatic tumor fibrosis and promotes nanoparticle delivery.
Article Type: regular article
Corresponding Authors: Gail Matters
Authors:
Thomas Abraham, Michael Armold, Christopher McGovern, John Harms, Matthew Darok, Christopher Gigliotti, Bernadette Adair, Jennifer Gray, Deborah F. Kelly, James Hansell Adair, Gail Matters
In this study, the authors have shown the effect of proglumide (CCK2R antagonist) treatment on perfusion using the delivery of a nano-encapsulated imaging agent using Kras/ Pdx-Cre (KC) mice. They have also shown that the ability of proglumide to decrease tumor fibrosis was tested using both an orthotopic human xenograft model and a murine PDAC tumor model.
The authors concluded that CCK2R antagonists can improve the delivery of nano-capsulated therapeutics or imaging agents to pancreatic tumors.
The experimental works are nicely performed, and the conclusions are justified and stated well by the results.
I recommend this paper for the “Biomedicines” journal.
Author Response
Thank-you for your support of this work.